# Evaluating the Efficiency of Municipal Solid Waste Collection Services in Developing Countries: The Case of Chile



**Jean Pierre Doussoulin [1,2,\*]** and **Cristian Colther [1]**

1    Economics Institute, Universidad Austral de Chile, Valdivia 5090000, Chile
2    Research Group on the Use of Panel Data in Economics (ERUDITE EA 437), Université Gustave Eiffel, F-77454 Marne-la-Vallée, France
\*    Correspondence: jean.doussoulin@uach.cl

**Abstract:** Due to the increasing volume of waste and the increasingly complex administration of its collection and disposal, solid waste management is quickly becoming a demanding issue for municipalities all over the world. Benchmarking the effectiveness of municipal solid waste management is critical for assessing municipalities' resource management performance and developing public policies for improvement. The main contribution of this article is an analysis of the efficiency of municipal collection services in Chile focusing in house solid waste. This study estimates the economic and technical efficiency using Stochastic Frontier Models for socio-economic, technical and human geography data from 2014 to 2019 for a sample of 280 municipalities, as well as an analysis of the internal and external factors that influence the efficiency levels shown by municipalities using an econometric model with 2017 socio-economic data. In addition, the spatial distribution of efficiency is investigated, with the Moran index used to identify clusters of towns to see if there is any spatial autocorrelation. The findings show that there are considerable disparities depending on whether the collection is private, public or mixed, and that rural municipalities are inefficient. The efficiency is not distributed evenly throughout space. The findings and recommendations of this study are intended to aid in the improvement of municipal and public policies relating to MSW management efficiency.

**Keywords:** waste; municipal solid waste management; production function; Stochastic Frontier Analysis (SFA); technical efficiency





## 1. Introduction

As a result of the COVID-19 pandemic, we can note the disappearance of questions and conversations related to climate change towards other issues such as waste management which have gained sanitary importance [1]. In the recent discussion of the pros and cons of the circular economy and waste management, including the controversial question of whether to emphasize the limitations of waste disposal, the literature suggests that even more attention is needed to assess the effectiveness of municipal solid waste (MSW) system.

From a circular economy perspective, municipal waste management is of great relevance because it is the primary means by which you can implement strategies to reduce and recycle, and designs that facilitate the efficient use of the goods and services generated by the economy over time [2]. Authors create methodologies aimed at promoting innovative sustainability services and business innovation [3,4] or, more broadly, tools capable of prioritizing sustainability issues in terms of environmental risks and opportunities about materiality analysis [5].

On the one hand, solid waste management is gradually becoming a challenging task for municipalities around the world due to increasing waste volume, varying waste structure, decreasing landfill sites and associated environmental risk [6]. In addition, there is a dilemma regarding the use of public resources, which must be used efficiently, prioritizing their use to increase the well-being of citizens; however, as cities are urbanizing

and becoming denser, municipalities must prioritize the use of the scarce financial resources they have at their disposal, allocating an increasing amount of money and personnel to manage collection and disposal services of waste [7].

On the other hand, this has generated a waste market, where depending on the country analyzed, different forms of management can be observed around public, mixed and private waste collection services. Several studies have shown that there is a direct relationship between the type of management and the degree of efficiency regarding waste management at the municipal level [8–10]. As an outcome, this study examines the technical and economic efficiency of municipal waste management.

The efficiency was assessed using Stochastic Frontier Models and municipal management data such as costs, as well as data from the outside municipalities such as the territory's human geography and rural components.

The implications of this work in relation to the fiscal aspect and the municipal budget, which are useful for policymakers, are related to stimulating processes of articulation to this decentralized waste management, at an appropriate subnational level, such as regional, provincial or according to territories with challenges. Financing elements of moderation and continuous improvement, as well as assisting small municipalities with funding for better collection and assimilation of best practices, all help to reduce the high costs involved. Another intriguing aspect is the sharing of disposal costs, the development of intermediate disposal and recycling stations, and the redistribution of surplus resources in solidarity to assist less efficient municipalities. It appears that autonomous management without adequate funding makes it difficult for municipalities to achieve higher levels of efficiency, and it would therefore be recommended to allocate resources not only for collection, as in Chile, but also for waste disposal and logistics, which are required for greater technical efficiency.

Therefore, the efficiency and effectiveness of waste management systems in cities depend on their capabilities that are based on utilizing economic efficiency and technological advantages to foster social mobilization and environmental integrity [11]. In rural locations, the problem of efficiency is linked to waste management. This is due to the municipality's administrative and strategic disarray, which is linked to an insecure and inequitable pricing structure [6].

However, different municipalities (urban or rural) assign different levels of priority to waste management issues, and therefore different forms of management can be expected at the municipal level [12]. Significant resources are wasted, especially by inefficient organizations, due to inadequate administrative procedures and mismanagement [13], lack of knowledge of local authorities regarding proper management, or lack of technical knowledge and skills [14], so that competencies and abilities affect efficiency despite the economic resources and performance of assigned personnel.

Effective management of municipal waste services is necessary to reduce costs, provide better quality services, reduce tariffs for citizens, and meet the requirements outlined by the international organism [15]. In this context, various questions naturally arise related to the efficiency of municipalities to manage the waste collection. As cities become more urbanized and denser, are collection services more efficient? Does the management method used to collect waste affect the efficiency of municipalities? Is there a role for spatial configuration in the efficiency of municipalities?

These questions will be answered by analyzing the municipal waste collection system in Chile, which can be considered an interesting case due to its characteristics as an emerging economy in the Latin American context, which has undergone processes of urbanization and densification of its cities, but at the same time coexists with territories where rurality predominates. In addition, different types of municipalities coexist depending on their size and the level of financial resources available to meet the needs of their territory.

Furthermore, due to its commercial openness and economic development, its population assimilates every day the lifestyle of developed countries, causing a sustained increase in the production of waste [16]. Finally, because it is a market-oriented economy, a waste

market and different types of management that municipalities use can be observed in more developed urban centers, ranging from self-management of services, where municipalities hire personnel and have infrastructure and machinery to provide the service to the community, a mixed regime in which the municipalities partially outsource them, facilitating the infrastructure and machinery and leaving the management of the service and hiring of personnel to a private party, or completely outsourcing the service, when a private provides the infrastructure, machinery and personnel, fully managing the waste collection service.

The objectives of this study are: Firstly, to develop an MSWM framework. Secondly, to examine the impact that the type of management of waste collection services can have on the degree of efficiency of municipalities. Finally, to identify the spatial differences between the territories, according to their degree of efficiency and the factors which may affect it, and explore urban or rural variations in the efficiency of MSW collection in the northern, central and southern regions of Chile.

The study's main contribution is in three areas: Firstly, it provides new information on Chile, a Latin American country where the layout of the municipal waste collection and disposal system is becoming increasingly complex. Secondly, it provides background information on the impact that management style, as well as the degree of rurality, may have on municipal efficiency. Thirdly, it presents preliminary evidence on the impact of spatial distribution on municipal efficiency.

The significance of this article is that it addresses the criticisms on municipal activities by considering that the management of MSW problems is limited due to budget and resource constraints. The increased waste disposal processes justify the need for more effective municipal policies emphasizing sustainable and efficient MSW collection strategies in Latin American countries (LAC).

These findings have policy implications and can inform the related government departments on how to formulate appropriate policies to improve collection efficiency. Indeed, new information and expertise generated by research can help and support policy-making by providing reliable background information, generating the knowledge and tools needed in Chile and supporting policymakers and citizens' activities.

The rest of this article is organized as follows. Section 2 presents a brief review of the literature that has addressed this issue at a general level and in a particular way for the Chilean case. Section 3 shows the methodology followed in the study and the source of the information used. Section 4 presents the main results of the investigation. Section 5 presents the main conclusions and final comments.

## 2. Literature Review

### 2.1. Municipal Solid Waste Management

The study of the efficiency of municipalities in the collection of household waste is interesting from various points of view. For example, from a social perspective, the efficient use of public resources is of utmost importance, given the limited financial resources that local governments usually have to meet the multiple needs of citizens [17]. Furthermore, as more and more resources are allocated to waste collection services, other important community services such as education and health will see their budgets cut.

In terms of the environment, the rising trend of urbanization and population growth, alongside growing concern about negative environmental consequences, has produced a difficult scenario for the management of household solid waste [18]. An efficiency study can reveal the degree of sustainable development available to a territory, insofar as it develops waste management and collection system in a circular economy approach [2], where there are strategies of waste reduction and recycling, efficient use of energy, and proper waste disposal to avoid affecting the natural environment and the health of citizens.

Municipal solid waste management (MSWM) is defined as the complex procedures comprising waste collection routes, transfer station locations, treatment and energy recovery strategies and waste treatment techniques that are the primary means to achieve these

objectives, including human health and environmental protection, and compliance with social and regulatory standards [2,19].

From 1965 to 2016, more than 100 articles were published on efficiency in waste management [20–22]. Taking this framework into account, several studies have measured and explained the cost-effectiveness of waste management services in municipalities and local governments and can be grouped according to the statistical method used to measure efficiency.

The first group of articles is based on parametric methods. Some of these papers investigated the effect of changing environmental variables on increasing waste management costs rather than profitability [23–25]. There are many such references in the literature including the following. Fernández-Aracil [26] used a database of municipalities in Spain to conduct a cost analysis of waste management. Wilson and Game [27] analyzed the impact of the Compulsory Competitive Tendering (CCT) on Garbage Collection Services. Other authors are currently carrying on with this analysis of waste management costs related to compliance and separate collection [28,29] and organized crime [30].

Parametric methods have been used to evaluate the efficiency in the management of other services in other issues. Lampe and Hilgers [31] conducted a bibliographic review and examined the increase in scholarship on efficiency methods in the management of an organization from an energy perspective, focusing on two approaches (Data Envelopment Analysis (DEA) and Stochastic Frontier Analysis (SFA)). Mutz et al. [32] used a Bayesian stochastic frontier approach (B-SFA), which made use of data containing 1046 samples to analyze the scientific productivity in Austrian universities using data from the Austrian Science Fund (FWF).

The second group of articles refers to studies in which non-parametric methods are used. These articles aimed to analyze the efficiency of waste management using the non-parametric data envelopment analysis (DEA) method. Marques et al. [33] used a non-parametric method to analyze the efficiency of Portuguese recycling companies and argue that the lack of incentives is an important reason for the poor performance of Portuguese recycling systems. Guerrini et al. [34] studied the performance of waste collection services by analyzing their efficiency, analyzing 40 municipalities in the province of Verona in Italy. The main results show that the integration of the collection services of small municipalities does not produce a significant increase in efficiency.

The authors also argue that to increase efficiency, waste collection strategies should be adopted targeting non-residential customers' waste. As shown by Fan et al. [7], small and medium-sized enterprises engaged in manufacturing (tertiary industry) or retail trade can account for a significant percentage of waste to be collected in certain municipalities. Worthington and Dollery [35] used a DEA method to analyze the technical efficiency of household waste management in 103 New South Wales municipalities in Australia. The results of the article indicate that the lack of efficiency in the management of urban waste is largely the result of the congestion caused by the high population density. Molinos-Senante et al. [36] studied the efficiency of the role of the private sector in the management of water and sewerage in Chile by applying a DEA model. The results indicate that the efficiency varies depending on the use of conventional or double-bootstrap models. Another relevant aspect may be the Deposit-Refund System (DRS) [37]. Calabrese et al. [38] investigated the operational modes and cost burdens of ten European DRS. This study demonstrates that DRS is one of the most effective systems for collecting one-way beverage packaging and that it is relevant in Europe's recycling and circular economy public policies.

García-Sánchez [39] used a method consisting of three steps: Firstly, the author initially applied a DEA to analyze the efficiency in Spanish municipalities. Secondly, he identified the demographic and socioeconomic factors that can explain this efficiency, which were used to explain the difference in efficiency through a Tobit regression. In the end, he implemented a second DEA model, which analyzed the variables of the Tobit regression. Thirdly, the author concludes that municipalities can reduce the resources used in providing the garbage management service by 8 percent% through an improvement in management.

After analyzing these two groups of articles, the authors suggest that regardless of the analysis method used, there are territorial and local factors that affect efficiency. Some of these factors can be controlled and managed through the services provided by a company or the government. Others cannot be controlled and depend mainly on the economic, environmental, social or health context in which the service operates.

Fusco and Allegrini [40] argued that the factors which are most frequently used by scholars are the size, density or age of the population, tourist flow, per capita income, private or public waste recovery service and the government's political orientation. Geys and Moesen [41] classified municipalities to analyze efficiency according to their agricultural, residential, industrial, touristic and urban characteristics. Passarini et al. [42] analyzed local government according to altitude and population density and Rogge and De Jaeger [24] divided municipalities into residential and rural.

After carrying out a literature review and to our knowledge, no published study has taken into account the efficiency of municipal governments from a spatial perspective considering the following key variables in the efficiency function: the volume of MSW collection services (RSD), the cost of MSW collection services (SRV) and the number of people engaging in the MSW collection services (RECOL). The following factors influence efficiency in this study: rural (RUR), population (POP), population density (DP) and daily waste production per inhabitant (PPC).

### 2.2. Waste Management System in Chile

Like other LAC, Chile is making an effort to modernize waste management through the modernization of waste regulations proposed by the OECD [43] and the creation and articulation of a circular economy roadmap. These new requirements highlight the opportunities to improve municipal waste management. In LAC, the management of MSW is done by the municipal governments, which often use property taxes to finance trash management costs. Indeed, the waste collection, recovery and disposal cost from 2002 to 2020 rose by 42% [44].

Chile produced 49.9 million tons of non-hazardous industrial waste, corresponding to 65.7% of the manufacturing sector, 9.7% of the mining sector and 6% of the energy sector [45]. The amount of MSW produced increased from approximately 294.6 kg/(cap) in 2000 to approximately 439.7 kg/(cap) in 2017, with a total generation of 7.5 million tons in 2017, where the Metropolitan Region accounts for 41.8%, followed by the Valparaiso Region with 11.5%, and 97% of MSW is disposed of, with only 3% collected for recovery [46]. In the country, the treatment of MSW is mainly limited to final disposal, without any kind of separation, composting and generation of energy from waste [47]. There is a strong informal waste management activity, which includes some type of waste recycling or reuse of electronic parts [48]. The country operates 38 sanitary landfills, 52 non-regulated landfills, and 38 dumpsites. The metropolitan Region of Santiago has three landfills producing 43% of the national solid waste, followed by the Valparaiso Region with 11.5% and the BíoBío Region with 7.5%, which generate greenhouse gases, contributing to climate change [46,49,50].

The following authors have made efforts to improve waste management in Chile. Weinstein [51] analyzed the waste-to-energy (WTE) ratio for Santiago, Chile using a Cost-Benefit Analysis. Melgarejo and Molinos-Senante [52] analyzed the eco-productivity change in the MSW management services. The authors all noted the scarcity of studies allowing the analysis of the efficiency of municipal waste management in Chile, which could help to face the challenges of the circular economy.

### 3. Methodology and Data

#### 3.1. Sample and Data Description

In this study, a data panel was constructed for the 2014–2019 period with a sample of 280 municipalities of the 345 existing ones in the country, representing 81% of the total municipalities in the country and 97.7% of the population. The data for this study

is from the Chilean Municipal Information System SINIM [53] and the data waste of SUBDERE [54]. The SINIM system is a management tool that consolidates a group of variables and the statistical data of municipalities, and is the main source of information for municipal issues as it includes information on management, finance, human resources and municipal characterization [55]. There were no significant changes in regulatory standards or technological advancements in the solid waste collection service, which was carried out by decentralized management of municipalities using very similar collection techniques and machinery during the time period under consideration.

Following previous studies on the efficiency of waste service management [7,15] the annual tons of garbage collected by municipalities (y) were included as input.

The cost of waste management municipalities (x1) and the number of workers dedicated to the collection of waste (x2) were included as inputs as it has always been the case in this type of study. The cost included costs for collecting and transporting the unsorted waste, storing it in transitory stations (in some cases) and finally disposing it at a final disposal site (landfill) of the unsorted waste. We control the degree of efficiency using the following variables: a geographic variable that considers whether the territory is predominantly rural or urban (x3), a variable related to the type of management used by municipalities for the collection of waste, which in the case of Chile are three, public management (x4), which is when the municipality is in charge of waste collection with workers, infrastructure and machinery for it; a mixed one (x5), where the municipality provides the infrastructure and machinery to a private party for the management of resources and hiring of staff; and a completely private one (x6), where the municipality outsources the provision of the service to a private party through public bidding.

In selecting the potential exogenous variables affecting efficiency, we took into account the features of the MSW sector, the data available to municipalities and the extant literature (e.g., [7,15,34]). As mentioned by Romano and Molinos-Senante [15], a consensus has not been reached in the existing literature on exogenous and environmental variables that can have an impact on the degree of efficiency that municipalities can have managing their waste, and contradictory results can also be observed, and therefore more research needs to be done to clarify this issue.

In this sense, we wish to contribute to the usual selection of variables to be considered as efficiency influencing variables, separating them into two dimensions: those linked to the characteristics of the municipalities (internal factors) and characteristics of the waste producers (environmental factors), as explained at the Table 1. Internal factors include organizational size, availability of resources and dependence on external resources for waste management [56]. Environmental factors, considering the characteristics of the families and companies generating the waste, which are the environment that municipalities face to provide the service, are linked to demographic and educational characteristics that are usually considered in the literature, and characteristics of the companies in the territory [7]. The variables used in this study are (i) number of employees A, (ii) ratio cost MSW/municipal budget, (iii) tax for waste collection C, (iv) budget per capita D, (v) population density E, (vi) gender ratio F, (vii) educational level G, (viii) number of enterprises H and added value of the enterprises per capita I, fully detailed in Appendix A, Table A1.

Finally, we looked at the spatial distribution of municipal efficiency, using the Moran index to determine the extent of spatial autocorrelation and see whether there are any spatial patterns or clusters of municipalities that have similar efficiency features. Moran's index is a statistical measure created by Alfred Pierce Moran that examines the spatial autocorrelation differences between nearest neighbor values, categorizing them as positive, negative or no spatial autocorrelation. A positive spatial autocorrelation exists when the values tend to cluster together; however, if the values are dispersed, the autocorrelation becomes negative, and if the values are scattered or randomly distributed, there is no spatial autocorrelation between the values evaluated [57].

**Table 1.** The statistical indicators of input-output variables in MSW collection services.

| Variable | Description | Observation Number | Average Value | Standard Deviation | Minimum Value | Maximum Value |
|---|---|---|---|---|---|---|
| Y | The volume of MSW collection services (tons) | 1681 | 26,070 | 39,417.0 | 30 | 360,450 |
| x1 | The investment of MSW collection services (thousands of Chilean Pesos) | 1681 | 1,153,809 | 1,865,729.0 | 178 | 14,765,504 |
| x2 | The number of people engaging in the MSW collection services (persons) | 1681 | 52.5 | 69.9 | 2 | 544 |
| x3 | Rurality (1: rural, 0: urban) | 360 | 0.24 | 0.43 | 0 | 1 |
| x4 | Management of MSW (1: Public, 0: no public) | 245 | 0.16 | 0.37 | 0 | 1 |
| x5 | Management of MSW (1: mixed, 0: no mixed) | 335 | 0.22 | 0.42 | 0 | 1 |
| x6 | Management of MSW (1: Private, 0: no private) | 870 | 0.61 | 0.49 | 0 | 1 |

*3.2. Methodology*

The proposed methodology was developed in three steps: First, the efficiency of the municipalities was estimated using a stochastics frontier model (SFM) proposed by Battese and Coelli [58], which estimates the inefficiency equation together with the stochastic frontier model in one step. Then, using cross-sectional data from 2017, an econometric model was developed that investigates the influencing elements of internal municipal management characteristics and external socioeconomic characteristics faced by municipalities that may affect the level of efficiency they display. Finally, we looked at the spatial distribution of municipal efficiency, using the Moran index [57] to determine the extent of spatial autocorrelation and see whether there are any spatial patterns or clusters of municipalities with comparable efficiency characteristics.

To determine the degree of efficiency of municipalities, an aggregate Cobb–Douglas production function in its logarithmic form was estimated using econometric panel data methods and through a stochastic boundary analysis. In the case of the production function model, the following functional form can be used (see Equation (1)).

$$\ln(y_{it}) = \ln f(X_{it}, \beta) + v_{it} - u_{it}, \ i = 1, 2, \ldots, 280; \ t = 1, 2, \ldots, 6 \tag{1}$$

where $y_{it}$ represents the volume of MSW collection service system in the $i$th municipality, $f(..)$ is the deterministic frontier output on the production possibility frontier, the maximum output with full efficiency; $X_{it}$ denotes the vector of inputs of the $i$th municipality at time $t$; $\beta$ represents a vector of parameters to be estimated; $v_{it} - u_{it}$ is a composite error structure with independent and normally distributed components; $v_{it}$ stands for the effect of random factors on output; and $u_{it}$ denotes the effect of technical inefficiencies on output [7].

The SF model is motivated by the theoretical idea that no economic agent can exceed an "ideal" efficiency frontier and that any deviation from this extreme represents individual deficiencies [59]. From a statistical point of view, this idea can be implemented by specifying a regression model characterized by a compound error term that includes the classical idiosyncratic term of disturbance, and the disturbance or error represented by inefficiency. Regardless of whether it is sectional or panel data, production or cost frontier, variable or invariable inefficiency, SF parametric models are generally estimated by methods based on maximum likelihood (ML) probability.

The SF model assumes that each municipality produces potentially less than what it can because of some degree of inefficiency. Specifically, a production function of the type

$$y_i = f(X_i, \beta)\varepsilon_i \tag{2}$$

where $\varepsilon_i$ is the level of efficiency for the municipality $i$ and must necessarily be in the interval (0, 1). If $\varepsilon_i$ is equal to 1, the municipality is achieving the optimum result with the resource incorporated in the production function $f(X_i, \beta)$. When $\varepsilon_i < 1$, the municipality is not taking full advantage of the $X_i$ inputs given the resources incorporated in the production function. Assuming that the output is strictly positive (i.e., $y_i > 0$), it is assumed that the degree of technical efficiency is strictly positive (i.e., $\varepsilon_i > 0$). It is also assumed that the output is subject to random shocks, which implies that

$$y_i = f(X_i, \beta)\varepsilon_i \exp(v_i) \tag{3}$$

Taking the natural logarithm on both sides

$$\ln(y_i) = \ln(f(X_i, \beta)) + \ln(\varepsilon_i) + v_i \tag{4}$$

Assuming that there are $k$ entries and that the production function is linear in logarithms, defining $u_i = -\ln(\varepsilon_i)$, one has to use the following Equation (1). Because $u_i$ is subtracted from $\ln(q_i)$, restricting $u_i \geq 0$ implies that $0 < \varepsilon_i \leq 1$, as specified in Equation (4) [60].

This paper applied the function proposed by Battese and Coelli [58] to analyze the efficiency of MSW collection services in Chile for explaining the determinants of technical efficiency via two stages; first estimate the stochastic frontier production function and determine the technical efficiency index, and then regress the technical efficiency index on the hypothesized variables to affect technical efficiency, to obtain an estimate of the degree of influence of these factors on technical efficiency. This methodology has been used in various studies (see [7]).

In order to estimate the factors that influence the efficiency of municipalities, the information for the year 2017 was considered due to the limitation of having panel data in the socioeconomic sphere for the period 2014–2019.

$$e_i = \delta_0 + \delta_1 A_i + \delta_2 B_i + \delta_3 C_i + \delta_4 D_i + \delta_5 E_i + \delta_6 F_i + \delta_7 G_i + \delta_8 H_i + \delta_9 I_i + \omega_i \tag{5}$$

where $e_i$, as mentioned above, is the efficiency of the MSW collection service in the $i$th municipality; $\delta_0$ signifies the parameters for the constant term; $\delta_1, \delta_2, \delta_3, \delta_4, \delta_5, \delta_6, \delta_7, \delta_8, \delta_9$, are used to denote the parameters of influence factor variables including the following log-transformed variables: The number of employees, management burden ratio, tax for waste collection, population density, income per capita, gender ratio, education level, number of enterprises and the added value of the enterprises per capita, respectively; $A_i, B_i, C_i, D_i, E_i, F_i, G_i, H_i, I_i$ represent the influencing factor variables of the MSW collection service in the $i$th municipality; $\omega_i$ is the random error term. The cross-sectional data econometric model considers correction for heteroskedasticity according to the specialized literature [61,62].

## 4. Results and Discussion

### 4.1. Main Results

Table 2 analyzes the efficiency behavior of five models. Model 1 includes all municipalities taking into account costs, personnel and the amount of waste that must be managed. Model 2 separates the municipalities that are rural from those that are not rural. Model 3 groups together the municipalities of public management. Model 4 includes those with public and private management and Model 5 represents municipalities with private management or a completely outsourced management to a private company.

Table 2 compares the efficiency of various types of municipalities represented in each model. In general, municipalities have a low level of efficiency in terms of waste management. Similar results were previously obtained by Fan et al. [7]. This variation in efficiency is related to the difficult management of waste by municipalities and the increase in its complexity every year due to an increase in the amount of waste generated and an

increase in population [63,64]. This justifies the fact that municipalities are increasingly economically and environmentally inefficient in Chile [65].

**Table 2.** SFA estimate panel data (2014–2019).

| Variable | Model 1 | Model 2 | Model 3 | Model 4 | Model 5 |
|---|---|---|---|---|---|
| $\beta_0$ | 7.56 (0.357) | 8.67 (0.318) | 7.40 (0.352) | 7.71 (0.342) | 7.62 (0.381) |
| $\beta_1$ | 0.15 (0.018) | 0.10 (0.015) | 0.15 (0.020) | 0.14 (0.018) | 0.14 (0.020) |
| $\beta_2$ | 0.43 (0.041) | 0.34 (0.040) | 0.44 (0.031) | 0.41 (0.036) | 0.42 (0.036) |
| Rural | | −1.49 (0.130) | | | |
| Public | | | 0.12 (0.073) | | |
| Mixed | | | | −0.27 (0.070) | |
| Private | | | | | 0.08 (0.070) |
| Time | −0.002 (0.002) | 0.002 (0.002) | −0.002 (0.002) | −0.001 (0.002) | −0.002 (0.002) |
| mean efficiency (cross-section, time periods) | 0.28 (285.6) | 0.29 (285.6) | 0.29 (285.6) | 0.28 (285.6) | 0.27 (285.6) |
| n$^o$ | 1681 | 1681 | 1681 | 1681 | 1681 |
| $\sigma^2$ | 3.66 *** | 3.29 *** | 3.53 *** | 3.73 *** | 3.76 *** |
| $\Gamma$ | 0.985 *** | 0.985 *** | 0.985 *** | 0.986 *** | 0.986 *** |
| Log likelihood value | −584.8 | −539.5 | −584.0 | −581.0 | −584.3 |

Signif. codes: *** *p*-value < 0.0001.

The economic implications of inefficiency are related to municipal costs and budgets [56]. Indeed, a larger budget is required each year to counteract this inefficiency and provide the best service to citizens. This budget increase encourages the intervention of the government, which grants a larger budget for the collection and management of waste, but also a larger budget to sensitize the population to reduce the generation of waste through waste separation systems. For instance, source-separated organic household waste, composting and reuse of textile waste, paper, plastic and glass and the generation of waste-to-energy [66]. The effectiveness of these programs will depend on the precariousness of the users and their food safety. A study in Brazil indicates that the degree of poverty influences the ability to apply waste recycling [67]. Considering this, the implementation of the circular economy programs depending on poverty can maintain a good level of efficiency in MSWM [68,69].

The literature demonstrates the well-known difficulties in waste management in pre-pandemic era in Chile, with negative externalities on public health, water resources, territorial biodiversity and increasing difficulty in managing hazardous waste, such as hospital waste, during the COVID-19 period [70]. Thus, the improvement or maintenance of efficiency is not only a matter of cost but also the collection and sending of waste to the landfill have generated a saturation of landfills in southern Chile.

Table 2 also indicates that rural municipalities are more inefficient due to the absence of an adequate waste collection and treatment system. These municipalities also present opportunities to implement waste recycling methods with the help of the state's waste [71]. This is based on the fact that its waste is very different from urban waste and with a high percentage of organic matter [72]. Therefore, wastes containing a high amount of organic matter can be transformed into compost for agricultural activity [73]. This reduces the cost of these types of products and the carbon footprint generated by rural economic activities which generally import their products to the place of use, generating greenhouse gases and an ecological footprint [74].

Regarding the collection system, the staff ($\beta_2$) is more relevant than it, which is related to financial expense ($\beta_1$). This result is in line with the results of the coefficients presented in the literature. Therefore, it would be desirable to increase spending on the education of personnel who participate in the collection and management of waste, understanding the positive response that the population has to these educational programs [75,76]. This will allow progress towards the design and social acceptance of more complex systems that include recycling, reuse or incineration.

As it has been studied and shown by the literature, public, mixed or private management generate different results [76,77]. It is relevant to mention that private administration is for large cities and public administration is concentrated within smaller municipalities [78]. For example, in the capital of the country, the main system is private, but in the regions and extremes of the country, the management system is municipal. Within public management, there are efficient cases that are associated with lower costs when management is public. When management is public, the municipality owns the landfill and therefore does not have to pay for on-site waste management. With the mixed system, the municipality must pay for the managed waste. If the system is completely private, the municipality must pay for the disposal and management of each ton of waste.

In public management, the municipality has the personnel to manage the waste and if it is being privatized, the costs for salaries to external personnel also rise, which will no longer be the minimum salary allowed by the country's legislation, but the salary that will best suit the business to maximize corporate profit.

It is necessary to mention that there has been a tendency to privatize services, based on the increasing complexity in waste management, related to collection logistics in large cities and the specificity of regulations and management of recycling and other greener alternatives [79]. Therefore, as waste management becomes more complex, the municipality outsources the service. Results show that mixed management is inefficient, but it responds to the reality of many municipalities and constitutes an intermediate solution to privatization [80]. There exists controversial evidence showing that the form of production, whether public or private in small cities, does not generate systematic differences in costs [81].

Figure 1 compares the efficiency of rural versus urban municipalities and different types of administration. The density distribution is higher and more effective in rural municipalities compared to urban ones. There are no great differences where public, private or mixed administrations are concerned.

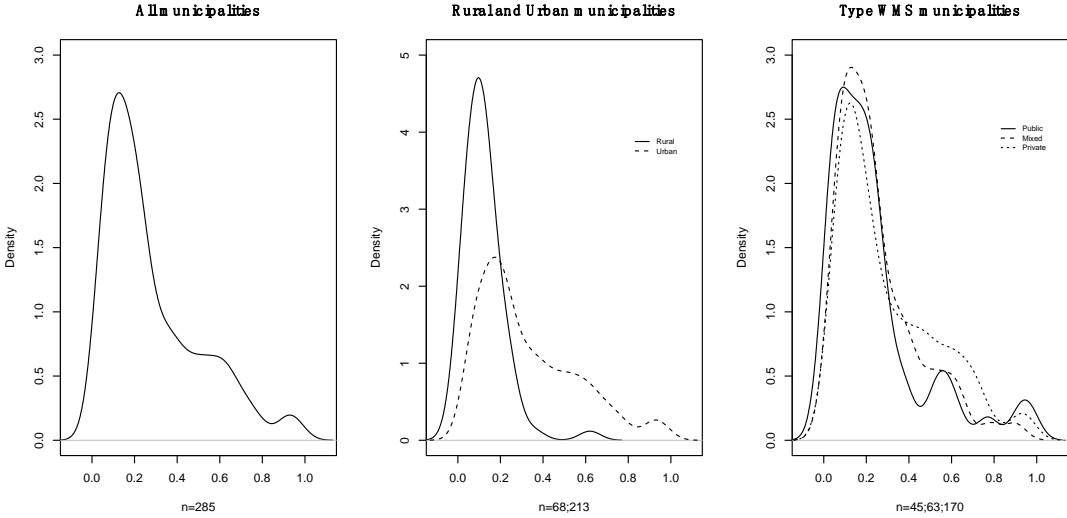

**Figure 1.** Kernel density of efficiency distributions for all municipalities ($n$ = 285), by type of municipality (rural $n$ = 68, urban $n$ = 2013) and type of collection service management (public $n$ = 45, mixed = 63, private = 170).

Table 3 indicates that the influencing variables were mostly positive. In this case, the influencing factors have been considered for the five efficiency models previously seen, in linear models expressed in terms of elasticities between the variables.

**Table 3.** SFA estimate panel data (2014–2019).

| Variable | Model 1 | Model 2 | Model 3 | Model 4 | Model 5 |
|---|---|---|---|---|---|
| $\beta_0$ | 7.56 (0.357) | 8.67 (0.318) | 7.40 (0.352) | 7.71 (0.342) | 7.62 (0.381) |
| $\beta_1$ | 0.15 (0.018) | 0.10 (0.015) | 0.15 (0.020) | 0.14 (0.018) | 0.14 (0.020) |
| $\beta_2$ | 0.43 (0.041) | 0.34 (0.040) | 0.44 (0.031) | 0.41 (0.036) | 0.42 (0.036) |
| Rural | | −1.49 (0.130) | | | |
| Public | | | 0.12 (0.073) | | |
| Mixed | | | | −0.27 (0.070) | |
| Private | | | | | 0.08 (0.070) |
| Time | −0.002 (0.002) | 0.002 (0.002) | −0.002 (0.002) | −0.001 (0.002) | −0.002 (0.002) |
| Mean efficiency (cross-section, time periods) | 0.28 (285.6) | 0.29 (285.6) | 0.29 (285.6) | 0.28 (285.6) | 0.27 (285.6) |
| n° | 1681 | 1681 | 1681 | 1681 | 1681 |
| $\sigma^2$ | 3.66 *** | 3.29 *** | 3.53 *** | 3.73 *** | 3.76 *** |
| Γ | 0.985 *** | 0.985 *** | 0.985 *** | 0.986 *** | 0.986 *** |
| Log likelihood value | −584.8 | −539.5 | −584.0 | −581.0 | −584.3 |

Signif. codes: *** *p*-value < 0.0001.

The models show a high concordance in the factors influencing the efficiency of the municipalities for their solid waste management. On the other hand, there is a similarity in the degree of influence that the factors have on efficiency, except for Model 2 where the values of variables *A* and *C* are inverted; however, both variables turn out to be non-significant.

Regarding the factors, the models suggest that five of the nine variables considered affect efficiency, of which some coincide with the expected signs according to specialized literature and others in contrast. For example, the signs of variables *B* and *E* are significant and according to what was expected from their influence; on the other hand, variables *D*, *F* and *H* presented signs opposite to what was expected.

In addition, the results show that the influencing factors have very different weights inefficiency, and therefore they can be sorted in order from the most to the least important: *F*, *E*, *H*, *B* and *D*. According to our methodological proposal, of the internal variables that affect efficiency, *B*, *E* and *D* are counted; and in the case of environmental factors, the variables *F* and *H*.

The population density (*D*) and the waste management burden (*B*) present the expected signs and are factors that influence the efficiency of the municipalities. We can say that increasing the population density increases the efficiency of the municipality. The effect of population density is in line with the literature studied. Nevertheless, results in the literature are controversial for this variable and there is no conclusive evidence. For example, Fan's results indicate that increasing population density decreases efficiency [7]. Guerrini et al. [34] and Simões and Marques [82] also complement these controversial results related to the influence of population density on municipal efficiency. These authors also argue that there might be a complex relationship between density and the type of collection. The high density allows the collection of a large amount of waste within a very short period and with low costs, but this requires complex management and therefore a

poorly managed high density can lead to a high level of inefficiency. In the case of Chile, the relationship between efficiency and density is positive when the municipalities present a high degree of organizational development. Therefore, municipalities with a high density correspond to municipalities with high organizational development. Increasing municipalities can manage more complex waste collection strategies. It is relevant to recognize that the incidence density is very low on efficiency. For example, a 1% increase in density implies a 0.042% increase in efficiency.

Variable *B* represents the burden of the cost of waste management for a municipality, where municipalities with greater financial capacity can develop management and structural elements that can lead to more efficient use of resources. In our case, it can be seen that by reducing the load by 1% this generates an increase of 0.115%. However, this variable has not been explored in other studies, so we cannot compare its level of influence in other contexts.

In regards to the per capita budget (*D*), the models show that it has a negative influence, and this may be due to the fact that as the municipalities have more resources, perhaps the management of household waste is not a priority for better use of the resources, and there is an incentive to neglect its management. This effect can be amplified in the case of outsourcing the service, where it becomes a monopoly market, with little incentive for better management of waste collection, and in the case of Chilean municipalities that are rural and therefore generally small.

The cost in these municipalities has an important value in their budget and has a negative influence on efficiency. There is international evidence that mentions the important and often unfunded waste collection system in undeveloped countries [83]. In Chile, there are a variety of costs relative to the wealth of the municipality. For example, there are municipalities with high income where the cost of the collection service in the budget is not very important versus municipalities with medium income where the costs of these services are relevant in their budget. Thus, small-income municipalities have lower waste management costs since they don't produce large amounts of waste [81]. It can also be argued that increasing the revenues of the municipalities makes them less efficient. When a municipality has more resources, it is less careful in managing them.

In the external influencing factors, only *F* and *H* were significant, in order of importance. In the case of the gender ratio, the result is contrary to what was expected and this may be due to the particular characteristic of Chile, where the female population has increased in recent years on a huge part of the territory; however, due to the characteristics of the labor markets, the more urban territories tend to have a more balanced ratio and be more efficient in their management.

Business density is a factor that positively affects efficiency, which is contrary to our initial expectations. This result may again be as a result of the important differences between urban versus rural municipalities because the urban municipalities have higher company densities, and although they generate more waste, these generally take care of their disposal in a particular way, so a higher density may imply a lower demand for garbage collection services from a segment of the population in certain industrial neighborhoods. In this case, a 1% increase in the number of companies implies a 0.13% increase in the efficiency presented by the municipalities.

Finally, we can say regarding the variables that were not significant that, in regards to the number of municipal employees, this result may be due to the fact that starting from a certain size, increasing the number of civil service plants does not imply an improvement in the waste management system, if this increase is not directly related to waste management, and therefore it may be the case that municipalities with different degrees of efficiency have similar numbers of civil service plants. In the case of taxes linked to garbage collection, Chile has a particular result, since there are indirect taxes, linked to contributions from homes, and therefore collected mainly in urban areas. Territories have very different population densities and similar collection levels as a result of the tax exemptions for social housing and nearby rural areas. In the case of the education variable, the variable is not significant,

and this may be due to the fact that territories with different levels of efficiency have similar levels of the population with a high educational level. In the case of Chile, this effect may be important due to the popularization of higher education as experienced over the last decade. In the case of company sales, this variable is not significant because, in Chile, the concentration of business activities obeys a logic of administrative centralization where they are concentrated in regional and provincial capitals often preferred by companies with a greater volume of sales.

### 4.2. Results by Spatial Distribution Municipality

Figure 2 indicates that efficiency is related to the territorial issue. When studying the distribution, we can see that efficiency is not randomly distributed in space, with clusters of municipalities having similar efficiencies. The Moran index of 0.18 indicates that there is a spatial self-correlation. This indicates that there are groupings of municipalities with similar efficiencies. The most efficient sectors in the metropolitan region are separated from the most inefficient ones. It is necessary to mention that we are not comparing the quality of the service, but the effectiveness of the budget management of the service. The municipalities in the center of the metropolitan region have similar densities, but different efficiencies. Therefore, we cannot associate these factors directly at the territorial level, although there are studies that associate efficiency on a territorial scale, such as [84].

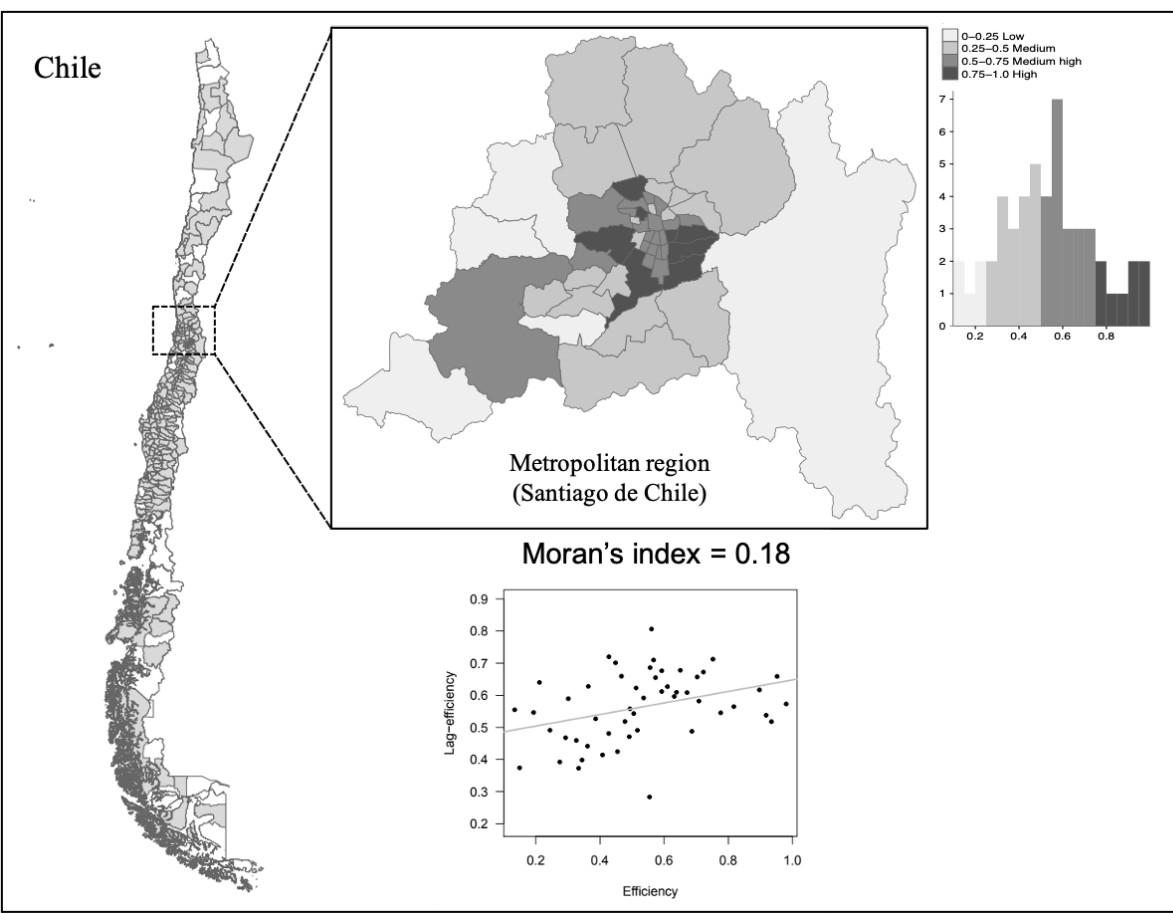

**Figure 2.** Spatial distribution municipality efficiency.

## 5. Implications

### 5.1. Theoretical Implications

This article has contributed to the literature through the study of types of efficiency in municipal waste management (public, private or mixed), the implications of the collection through taxes and the cost ratio of MSW/municipal budget.

There are studies like Fan et al. [7] that make no difference in the type of administration, assuming that it is completely public. Therefore, our article gives the possibility of studying efficiency by type of management: public, private or mixed. Our findings show significant differences in technical efficiency achieved depending on the type of waste collection management in a municipality in a country where private companies are permitted to participate in service provision.

### 5.2. Management Implications

In the results referring to the spatial distribution, the Moran index indicates that the efficiency is not randomly distributed in space. Therefore, municipalities tend to be grouped according to their level of efficiency. From a management implications point of view, in the Chilean case, it makes sense to carry out a concerted waste collection management between municipalities. One criterion that could be used is efficiency. Figure 2 showed that the municipalities of Santiago have similarities and can be arranged into groups. These similarities would allow for an organization oriented towards concerted waste management. It would be interesting to explore the existence of economies of scale organizational processes and the management of border points between municipalities, which are generally more difficult to manage. This would increase the effectiveness of public urban waste management policies.

## 6. Conclusions

The efficiency of municipalities in the management of household solid waste, as well as the internal and external factors that influence their performance, were investigated in this study. For the period 2014–2019, the findings revealed significant differences in terms of efficiency levels presented by municipalities in Chile, as well as differences in efficiency levels presented by urban versus rural municipalities and differences in public versus mixed administration solid waste management.

The population density and waste management load show the predicted indicators, and these are some of the characteristics that most influence the efficiency of Chilean cities, according to the analysis of the factors influencing municipal efficiency.

Efficiency is not randomly distributed in space; therefore, there are groupings according to levels of efficiency. There are groupings of municipalities with high levels of efficiency and groupings of municipalities with low levels of efficiency.

Our study on municipal efficiency constitutes a first approximation and can be used as a guide to compare efficiencies, but both cases require more in-depth and detailed study to determine the factors influencing efficiency. These factors are related to the organization and use of resources in each municipality and their distribution on the territory.

Efficiency can be used as a criterion for targeting public policy intervention. This allows the most inefficient to intervene, helping them if the inefficiency is due to scarce resources, as in this case, the government can invest resources to improve waste management. If the inefficiency is a result of a deficiency in management or an organizational problem, it is possible to transfer best practices to improve efficiency, which is observed in efficient municipalities or from international experience in MSW.

This work also proposed some limitations and generated ideas for future research. In recent years, there have been advances in SFA. The Bayesian approach is currently emerging. This adds the stochastic element and treats it as a Bayesian random element [85].

Therefore, it could be estimated if there are significant differences when studying efficiency with other methods. There is a variety of SFA which treats the random part in

the error as a normal, asymmetric distribution or use the maximum likelihood method [86]. Critics argue that the ranking varies according to the SFA used.

Future examinations to analyze the factors that affect efficiency in depth are required. Problems may have their origin in a deficient organization of the collection and logistics of waste or in a few resources. It would also be interesting to know the impact of recycling efficiency in the Chilean case. In European countries such as France, recycling has already been implemented and the cost savings of municipalities are measured by generating recycling habits in people. Recycling reduces the sending of waste to the landfill and allows value to be generated. The existence of studies on the efficiency of urban and rural recycling in developing countries would help to create and guide public debate on issues such as the circular economy roadmap.

**Author Contributions:** C.C.: Methodology, Data construction, Estimation, Writing—Reviewing and Editing. J.P.D.: Review literature, Writing—Reviewing and Editing. All authors have read and agreed to the published version of the manuscript.

**Funding:** This research received no external funding.

**Institutional Review Board Statement:** Not applicable.

**Informed Consent Statement:** Not applicable.

**Data Availability Statement:** The data that support the findings of this study are available from the authors, upon reasonable request.

**Conflicts of Interest:** The authors declare no conflict of interest.

## Appendix A

**Table A1.** Full explanation of the variables.

| Variable | Explanation | Expected Sign in Result |
|---|---|---|
| (i) *Number of employees A* | One of the characteristics of the municipality that we will consider in the study is the number of employees working in the municipality, since various studies mention the positive impact of this factor on the efficiency of the municipalities. This is consistent with the hypothesis that a municipality with a larger number of staff has a greater chance of adequately managing the services it offers to the community than those with fewer employees. | (+) |
| (ii) *Ratio cost MSW/municipal budget B* | This variable considers the percentage that the cost of waste management services occupies in the general budget. In this case, the hypothesis can be raised that those municipalities that have the greatest number of financial resources can better cope with the waste collection services and those in which it becomes a "heavy burden to manage" will be less efficient. | (−) |
| (iii) *Tax for waste collection C* | This variable considers the resources that enter the municipality for cleaning rights, which are collected at the central level, via taxes on residential housing that consider a percentage dedicated to household waste collection services. In this case, they are additional resources that Chilean municipalities receive to be used exclusively for cleaning streets and collecting household waste. The hypothesis in this case is that those municipalities that have exclusive additional resources to manage waste will be more efficient than those that do not have them. | (+) |

**Table A1.** *Cont.*

| Variable | Explanation | Expected Sign in Result |
|---|---|---|
| (iv) *Budget per capita D* | In this case, we consider the per capita budget of the municipality. Our hypothesis is related to the fact that a municipality with a greater availability of resources can develop strategies for more efficient collection methods and implement actions to reduce the waste generated. | (+) |
| (v) *Population density E* | This variable considers the number of people per square kilometer that the municipality must manage. In this case, the hypothesis is that those municipalities that must deal with densely populated territories should develop economies of scale that facilitate efficiency in the waste collection service and the possibility of implementing complementary services. | (+) |
| (vi) *Gender ratio F* | This variable considers is the percentage of women with respect to the total population. The hypothesis is that a municipality where there is a greater number of women may have a better performance because various studies have shown that women have a greater sensitivity with respect to better use of products, and therefore generate less waste. | (+) |
| (vii) *Education level G* | The educational level can be a factor that affects the degree of efficiency of the municipalities, since it can favor the consumption of reusable products, recycling practices and better use of the products, thus leading to less waste generation. | (+) |
| (viii) *Number of companies H* | The number of companies can influence the generation of waste experienced by a territory, because in the production of goods and services they use inputs that inevitably generate waste, particularly if they do not have a circular economy approach in their production. | (−) |
| (ix) *Added value of the enterprse per capita I,* | In this case, the added value per capita generated by companies can be a positive factor, because those territories with a higher level of value generation may have resources to improve their processes of production and optimize the use of their inputs to reduce waste generation and avoid production losses. | (+) |

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
