# Peer review of "Evaluating the Efficiency of Municipal Solid Waste Collection Services in Developing Countries: The Case of Chile"

_sustainability, doi:10.3390/su142315887_

Round 1

Reviewer 1 Report

This interesting paper deals with the issue of the efficiency of municipal solid waste management in Chile. There are still several issues to be solved before the paper can be considered ready for publication.

1-      Authors should explain how efficiency is measured in the introduction. Since efficiency is the main concept of the work, the reader should understand how it is measured from the beginning of the article.

2-      In the introduction, second paragraph (lines 35-38), the authors should improve the paper by citing articles that develop methodologies aimed at promoting innovative services in the field of sustainability or, more generally, tools capable of prioritising sustainability issues in terms of environmental risks and opportunities (materiality analysis for example). I suggest the authors utilize the following articles:

Calabrese, A., Forte, G., & Ghiron, N. L. (2018). Fostering sustainability-oriented service innovation (SOSI) through business model renewal: The SOSI tool. Journal of Cleaner Production, 201, 783-791.

Calabrese, A., Costa, R., Levialdi Ghiron, N., & Menichini, T. (2019). Materiality analysis in sustainability reporting: a tool for directing corporate sustainability towards emerging economic, environmental and social opportunities.

França, C. L., Broman, G., Robert, K. H., Basile, G., & Trygg, L. (2017). An approach to business model innovation and design for strategic sustainable development. Journal of Cleaner Production, 140, 155-166.

3-      The literature analysis about municipal solid waste management is dated, considering only the papers up to 2016. I suggest rewriting this paragraph, and inserting more updated papers, which consider multiple factors that influence the efficiency of municipal solid waste management, such as:

Barchiesi, M. A., Costa, R., & Di Pillo, F. (2022). The Link between the Compliance with Environmental Legislation on Separate Collection and the Municipal Solid Waste Costs. Sustainability, 14(9), 5661.

Di Pillo, F., Levialdi, N., & Marzano, R. (2022). Organized crime and waste management costs. Regional Studies, 1-13.

Fernández-Aracil, P., Ortuño-Padilla, A., & Melgarejo-Moreno, J. (2018). Factors related to municipal costs of waste collection service in Spain. Journal of Cleaner Production, 175, 553-560.

Musella, G., Agovino, M., Casaccia, M., & Crociata, A. (2019). Evaluating waste collection management: the case of macro-areas and municipalities in Italy. Environment, Development and Sustainability, 21(6), 2857-2889.

4-      Paragraph 5 should be deepened. What are the implications of this work relating to the fiscal aspect and the municipal budget useful for the policymakers? A discussion of this aspect would be welcomed.

5-      In paragraph 7, the authors refer to the interesting topic of the impact of recycling efficiency, which in developed countries is addressed through appropriate regulatory measures, such as deposit-refund systems. In this regard, the authors should include some works that deal with this issue, such as:

Calabrese, A., Costa, R., Ghiron, N. L., Menichini, T., Miscoli, V., & Tiburzi, L. (2021). Operating modes and cost burdens for the European deposit-refund systems: A systematic approach for their analysis and design. Journal of Cleaner Production, 288, 125600. 

Zhou, G., Gu, Y., Wu, Y., Gong, Y., Mu, X., Han, H., & Chang, T. (2020). A systematic review of the deposit-refund system for beverage packaging: Operating mode, key parameter and development trend. Journal of Cleaner Production, 251, 119660.

Author Response

Dear Reviewer, Thank you for your insightful and constructive advice. The changes are highlighted in yellow in the manuscript.

Response to the comments and suggestions of Reviewer#1

Reviewer #1: Notes on "Evaluating the efficiency of municipal solid waste collection services in developing countries: the case of Chile"

Comment 1:Authors should explain how efficiency is measured in the introduction. Since efficiency is the main concept of the work, the reader should understand how it is measured from the beginning of the article.

Reponse: Thank you for your valuable and constructive suggestions. Based on your suggestions, we explained how efficiency is measured between lines 57-58 of the manuscript.

Comment 2:In the introduction, second paragraph (lines 35-38), the authors should improve the paper by citing articles that develop methodologies aimed at promoting innovative services in the field of sustainability or, more generally, tools capable of prioritising sustainability issues in terms of environmental risks and opportunities (materiality analysis for example). I suggest the authors utilize the following articles...

Reponse: Thank you for your constructive suggestions. In the new version, we included and cited recommended articles (between lines 40-43 of the manuscript).

Comment 3:The literature analysis about municipal solid waste management is dated, considering only the papers up to 2016. I suggest rewriting this paragraph, and inserting more updated papers, which consider multiple factors that influence the efficiency of municipal solid waste management, such as..

Reponse: Thank you very much for your careful and insightful comments. Based on your suggestions, we have added papers between lines 166-173.

Comment 4:Paragraph 5 should be deepened. What are the implications of this work relating to the fiscal aspect and the municipal budget useful for the policymakers? A discussion of this aspect would be welcomed.

Reponse: Thank you for your valuable and constructive suggestions. Based on your suggestions. We include the implications of this work relating to the fiscal aspect and the municipal budget for the policymakers (between lines 62-74 of the manuscript).

Comment 5:In paragraph 7, the authors refer to the interesting topic of the impact of recycling efficiency, which in developed countries is addressed through appropriate regulatory measures, such as deposit-refund systems. In this regard, the authors should include some works that deal with this issue, such as…

Reponse: Thank you for your valuable and constructive suggestions. Based on your suggestions, we modified the manuscript in two aspects:

Firstly, we completely agree with your comments and suggestions about the introduction. The revised details are reported between lines 203-207 of the manuscript.

Secondly, in the new version, we introduce eco-productivity concept in 2.2 section on lines 258-259.

Reviewer 2 Report

Evaluating the efficiency of municipal solid waste collection services in developing countries: the case of Chile

This study describes an analytical evaluation for the data related to the efficiency of the solid waste collection services for 280 municipalities in Chile using Stochastic Frontier Models. The evaluated data was for the period of six year starting from 2014 to 2019. The study comprised also an investigation for the internal and external governing parameters on the efficiency level of service. The paper is an interesting and well written and informative. The methodology was organized in a logical way and the results have the sufficient potential to be used in the practice applications especially for those related to the service efficiency of the rural municipalities. I recommend accepting this paper to be published in the Sustainability Journal.

Author Response

Dear Reviewer, Thank you for your insightful and constructive advice. The changes are highlighted in yellow in the manuscript.

Response to the comments and suggestions of Reviewer#2

Reviewer #2: This study describes an analytical evaluation for the data related to the efficiency of the solid waste collection services for 280 municipalities in Chile using Stochastic Frontier Models. The evaluated data was for the period of six year starting from 2014 to 2019. The study comprised also an investigation for the internal and external governing parameters on the efficiency level of service. The paper is an interesting and well written and informative. The methodology was organized in a logical way and the results have the sufficient potential to be used in the practice applications especially for those related to the service efficiency of the rural municipalities.

Comment 1: I recommend accepting this paper to be published in the Sustainability Journal.

Reponse 1:Thank you for your comments. Your positive feedback means a lot to us.

Reviewer 3 Report

This research work refers to the efficiency of municipal solid waste collection services in Chile, a developing country. The paper is well structured and easy to understand. The problem with the MSWM is very important and has occupied many researchers. 

Τhere are some minor changes that should be made to be published : 

1. In line 551, on conclusions, the researchers wrote " The efficiency of municipalities in the management of household solid waste, as well as the internal and external factors that influence their performance, were investigated in this study." But in the abstract is not clear that this research is focusing on household solid waste. Please change a little bit the abstract.

2. In line 72 " we wish to answer " the term is not probationary. Please change it

3. In line 175 "The authors also argue that to increase efficiency, waste collection strategies should be adopted targeting non-residential customers' waste. The authors also argue that to increase efficiency, waste collection strategies should be adopted targeting non-residential customers' waste." Why do the authors argue about that? Do they have proof? They did a research or another researcher did? Please explain

Author Response

Dear Reviewer, Thank you for your insightful and constructive advice. The changes are highlighted in yellow in the manuscript.

Response to the comments and suggestions of Reviewer#3

Reviewer #3: Evaluating the efficiency of municipal solid waste collection services in developing countries: the case of Chile? Recommendation – Minor Revision

Comment 1:In line 551, on conclusions, the researchers wrote " The efficiency of municipalities in the management of household solid waste, as well as the internal and external factors that influence their performance, were investigated in this study." But in the abstract is not clear that this research is focusing on household solid waste. Please change a little bit the abstract.

Reponse: Thank you for your constructive suggestions. We include the suggestions on lines 14-15 of the manuscript.

Comment 2:In line 72 " we wish to answer " the term is not probationary. Please change it.

Reponse: Thanks for your valuable suggestions. We modified the manuscript on lines 96 and 97 of the manuscript.

Comment 3:In line 175 "The authors also argue that to increase efficiency, waste collection strategies should be adopted targeting non-residential customers' waste. The authors also argue that to increase efficiency, waste collection strategies should be adopted targeting non-residential customers' waste." Why do the authors argue about that? Do they have proof? They did a research or another researcher did? Please explain.

Reponse: Thank you for your valuable and constructive suggestions. We include the suggestions between lines 194-196.

Reviewer 4 Report

The manuscript, “Evaluating the efficiency of municipal solid waste collection services in devel-2 oping countries: the case of Chile,” fits the scope of the sustainability journal. However, the discussion in the result section is weak from an academic perspective, and the manuscript needs more meaningful conclusions. The manuscript often uses inappropriate language and needs grammar and word choice corrections.

1.     Line 116. The abbreviation of LAC must be defined in the first use.

2.     The authors used the term “efficiency” too often efficiency, For example, “The efficiency and effectiveness of waste management systems”, “the efficiency of municipalities to manage”, and “to improve collection efficiency”; however, it is not clear what efficiency indicates exactly in the context.

3.     In Line 227, “There is a strong informal waste management activity, which includes some type of waste recycling or reuse of electronic parts [42].” What is the contribution of the informal waste collection sector in the studied area? Does it affect the data set in use for this study?

4.     The authors should avoid the use of informal language in a technical manuscript. Line 280-282, “In order to estimate the factors that influence the efficiency of municipalities, let’s consider data corresponding to the year 2017 due to the limitation of having panel data in the socioeconomic sphere for the period 2014-2019.”

5.     There are many parameters that the authors or the other researchers considered in estimating waste collection efficiency. It is useful for the readers if the authors provide a table that summarizes the parameters in different categories and compares what is studied in this article and those in the literature.

6.     In this study, data was used from the 2014-2019 period. It must be clarified if there was any regulatory standard revision or technological improvement in the waste collection service. Therefore, the boundary of time and techniques on data should be guaranteed.

7.     In Line 358-59, “In general terms, municipalities are increasingly inefficient as waste management is concerned.”  What does mean by “increasingly inefficient” here?

8.     In line 377-379, “This situation is producing negative externalities on public health, water resources, territorial biodiversity and increasing the difficulty to manage hazardous waste such as hospital waste from COVID-19 [61].” The authors gave an example with “hospital waste from COVID-19” to address issues on the inefficiency of waste collection. However, this study used a data set between 2014 and 2019 before the COVID-19 pandemic broke out.

9.     In line 369-370. “organic in origin, composting and reuse of textile waste, paper, plastic, glass and the generation of waste-to-energy [57]” is an incomplete sentence.

10.  Figure 1. Distributions efficiencies municipalities needs units on the X and Y-axis and a clear indication of ns.

11.  Line 531-537, the authors need to describe the implication in more detail.

12.  This article analyzed the efficiency of MSW collection services in Chile using Stochastic Frontier Models for data. What are the recommendations of this study to improve the efficiency of the municipalities’ waste collection services? 

Author Response

Dear Reviewer, Thank you for your insightful and constructive advice. The changes are highlighted in yellow in the manuscript.

Response to the comments and suggestions of Reviewer#4

Reviewer #4: The manuscript, “Evaluating the efficiency of municipal solid waste collection services in devel-2 oping countries: the case of Chile,” fits the scope of the sustainability journal. However, the discussion in the result section is weak from an academic perspective, and the manuscript needs more meaningful conclusions. The manuscript often uses inappropriate language and needs grammar and word choice corrections.

Comment 1:Line 116. The abbreviation of LAC must be defined in the first use.

Reponse: Thank you for your comments. Based on your suggestions, the abbreviation was defined in line 130.

Comment 2:The authors used the term “efficiency” too often efficiency, For example, “The efficiency and effectiveness of waste management systems”, “the efficiency of municipalities to manage”, and “to improve collection efficiency”; however, it is not clear what efficiency indicates exactly in the context.

Reponse: Thank you for your constructive suggestions. In the new version, we added clarification about the efficiency in the abstract (between lines 15-17 of the manuscript) and in the introduction on lines 57 and 58 of the text.

Comment 3:In Line 227, “There is a strong informal waste management activity, which includes some type of waste recycling or reuse of electronic parts [42].” What is the contribution of the informal waste collection sector in the studied area? Does it affect the data set in use for this study?

Reponse: Thank you for your constructive suggestions. In the new version, we removed this line because it could lead to confusion.

Comment 4:The authors should avoid the use of informal language in a technical manuscript. Line 280-282, “In order to estimate the factors that influence the efficiency of municipalities, let’s consider data corresponding to the year 2017 due to the limitation of having panel data in the socioeconomic sphere for the period 2014-2019.”

Reponse: We'd like you to apologize for this oversight. In the new version, we modified the manuscript on lines 371 and 372.

Comment 5:There are many parameters that the authors or the other researchers considered in estimating waste collection efficiency. It is useful for the readers if the authors provide a table that summarizes the parameters in different categories and compares what is studied in this article and those in the literature.

Reponse: Thank you for your valuable and constructive suggestions. Previous research has compiled factors that can affect technical efficiency (for example, Fan et al), but in the type of study we conducted, technical efficiencies are not comparable between cases studied, because technical efficiency is measured between compared units.

Comment 6:In this study, data was used from the 2014-2019 period. It must be clarified if there was any regulatory standard revision or technological improvement in the waste collection service. Therefore, the boundary of time and techniques on data should be guaranteed.

Reponse: Thank you for your valuable and constructive suggestions. Based on your suggestions, we added some clarification between lines 272-275.

Comment 7:In Line 358-59, “In general terms, municipalities are increasingly inefficient as waste management is concerned.”  What does mean by “increasingly inefficient” here?

Reponse: Thank you very much for your careful and insightful comments. Based on your suggestions, we modified the text between lines 394-396.

Comment 8:In line 377-379, “This situation is producing negative externalities on public health, water resources, territorial biodiversity and increasing the difficulty to manage hazardous waste such as hospital waste from COVID-19 [61].” The authors gave an example with “hospital waste from COVID-19” to address issues on the inefficiency of waste collection. However, this study used a data set between 2014 and 2019 before the COVID-19 pandemic broke out.

Reponse: Thank you very much for your insightful comments. Based on your suggestions, we changed the text between lines 413-418.

Comment 9:In line 369-370. “organic in origin, composting and reuse of textile waste, paper, plastic, glass and the generation of waste-to-energy [57]” is an incomplete sentence.

Reponse: Thank you for your valuable and constructive suggestions. Based on your comments, we modified the manuscript on lines 406 and 407.

Comment 10:Figure 1. Distributions efficiencies municipalities needs units on the X and Y-axis and a clear indication of ns

Reponse: Thank you very much for your insightful comments. Based on your suggestions, we modified the manuscript between lines 463-465.

Comment 11:Line 531-537, the authors need to describe the implication in more detail.

Reponse: Thank you for your valuable and constructive suggestions. Based on your suggestions, we described the implication in more detail between lines 579-581 of the manuscript.

Comment 12:This article analyzed the efficiency of MSW collection services in Chile using Stochastic Frontier Models for data. What are the recommendations of this study to improve the efficiency of the municipalities’ waste collection services?

Reponse: Thank you for your comments. In the new version, the recommendation to improve efficiency is available between lines 256-259 and in point IX of the appendix section on page 17.

Reviewer 5 Report

Dear authors, please find my comments below: I hope this will assist in improving the quality of the study.

The abstract should also contain the studied country's name, "Chile." It also should indicate what type of data was used for analysis and what type of efficiency was calculated from that data (i.e., technical, economic, etc.).

Lines 96–105 should be part of the methodology section instead of the introduction.

Section "3.2. Sample and data description" should be placed before the methodology.

The section on results should be titled "Results and Discussion."

"7. Limitations and future direction of research" should be combined with the Conclusion section.

Author Response

Dear Reviewer, Thank you for your insightful and constructive advice. The changes are highlighted in yellow in the manuscript.

Response to the comments and suggestions of Reviewer#5

Reviewer #5: Evaluating the efficiency of municipal solid waste collection services in developing countries: the case of Chile?  

Comment 1:The abstract should also contain the studied country's name, "Chile." It also should indicate what type of data was used for analysis and what type of efficiency was calculated from that data (i.e., technical, economic, etc.).

Reponse: Thank you for your valuable and constructive suggestions. Based on your suggestions, we revised the manuscript between lines 14-17.

Comment 2:Lines 96–105 should be part of the methodology section instead of the introduction.

Reponse: Thank you for your constructive suggestions. Based on your suggestions, we revised the manuscript between lines 322-331.

Comment 3:Section "3.2. Sample and data description" should be placed before the methodology.

Reponse: Thanks for your valuable suggestions. We modified the manuscript between lines 264-320 of the manuscript.

Comment 4:The section on results should be titled "Results and Discussion."

Reponse: Done

Comment 5:Limitations and future direction of research" should be combined with the Conclusion section.

Reponse: Done

Round 2

Reviewer 1 Report

The authors improved the paper that is now ready for publishing